# Sinapic Acid Alleviates Oxidative Stress and Neuro-Inflammatory Changes in Sporadic Model of Alzheimer’s Disease in Rats

**DOI:** 10.3390/brainsci10120923

**Published:** 2020-11-30

**Authors:** Vandna Verma, Devendra Singh, Reeta KH

**Affiliations:** Department of Pharmacology, AIIMS, New Delhi 110029, India; drvandna89@gmail.com (V.V.); dvndr.singh@gmail.com (D.S.)

**Keywords:** Alzheimer’s disease, sinapic acid, ICV-STZ, choline acetyltransferase, neuro-inflammation, oxidative stress

## Abstract

The role of oxidative stress, neuro-inflammation and cholinergic dysfunction is already established in the development of Alzheimer’s disease (AD). Sinapic acid (SA), a hydroxylcinnamic acid derivative, has shown neuro-protective effects. The current study evaluates the neuro-protective potential of SA in intracerebroventricular streptozotocin (ICV-STZ) induced cognitive impairment in rats. Male Wistar rats were bilaterally injected with ICV-STZ. SA was administered intragastrically once daily for three weeks. Rats were divided into sham, ICV-STZ, STZ + SA (10 mg/kg), STZ + SA (20 mg/kg) and SA per se (20 mg/kg). Behavioral tests were assessed on day 0 and 21 days after STZ. Later, rats were sacrificed for biochemical parameters, pro-inflammatory cytokines, choline acetyltransferase (ChAT) expression and neuronal loss in the CA1 region of the hippocampus. The results showed that SA 20 mg/kg significantly (*p* < 0.05) improved cognitive impairment as assessed by Morris water maze and passive avoidance tests. SA 20 mg/kg reinstated the altered levels of GSH, MDA, TNF-α and IL-1β in the cortex and hippocampus. STZ-induced decreased expression of ChAT and neuronal loss were also significantly (*p* < 0.05) improved with SA. Our results showed that SA exhibits neuro-protection against ICV-STZ induced oxidative stress, neuro-inflammation, cholinergic dysfunction and neuronal loss, suggesting its potential in improving learning and memory in patients of AD.

## 1. Introduction

Approximately 80% of cases of dementia are linked to Alzheimer’s disease (AD), making it the major form of dementia [1]. According to the world Alzheimer’s report, the numbers of AD patients are expected to increase from 56 million in 2020 to 88 million by 2050. Thus, AD poses not only a great social impact, but would also result in a huge economic burden to healthcare worldwide [2]. Current treatments with tacrine, rivastigmine, galantamine, donepezil and memantine neither halt nor delay the progression of the disease, rather they only provide symptomatic relief [3]. The majority of the patients with AD have late onset called sporadic AD (>60 year), and less than 1% of patients have inherited mutations in genes that affect the processing of amyloid beta (Aβ) and these patients develop the disease at an early age (<60 year) called familial AD [4]. Several mechanisms are involved in neuronal degeneration in sporadic AD, including oxidative stress, cholinergic dysfunction, mitochondrial dysfunction, neuro-inflammation, apoptosis and genetic factors [5,6,7]. Consequently, these multiple factors may interact and exacerbate each other, leading to neuronal dysfunction and eventually cell death [8,9,10,11]. Therefore, there is an unmet need for developing novel therapies targeting these multiple pathways. 

Familial AD is mimicked using transgenic models whereas the more prevalent but late onset sporadic AD is mimicked credibly by streptozotocin (STZ) injections. Low dose intracerebroventricular (ICV) administration of STZ in rodents induces an insulin resistance-like brain state including behavioral, neurochemical and histological features resembling sporadic AD [12]. ICV-STZ produces neuronal damage through the production of reactive oxygen species, nitrogen species, neuro-inflammation, impaired insulin signaling, hyper-phosphorylation of tau and amyloid beta proteins [13,14,15,16]. Impaired brain energy metabolism reduces synthesis of acetyl coenzyme A and acetylcholine. Consequently, it interrupts the cholinergic transmission in neurons, ultimately leading to cholinergic deficits along with reduced choline acetyltransferase (ChAT) [9,17]. Neuro-inflammation along with oxidative stress plays an important role in the exacerbation and progression of AD by activation of astrocytes and microglia, releasing cytokines like IL-1, IL-6 and TNF-α, leading to Aβ generation and the development of AD [18,19].

Sinapic acid (SA), a natural herbal phenolic acid compound, is present in oranges, grapefruits, and cranberries or herbs like *Sinapis alba*, canola, mustard seed and rapeseed [20]. SA has been reported to be neuro-protective against amyloid beta induced AD and kainic acid-induced neuronal damage [21,22]. Furthermore, it prevented cognitive impairment in scopolamine induced amnesic mouse model [23]. In addition, SA has shown anti-inflammatory activity in carbon tetrachloride induced acute hepatic injury in rats [24]. Therefore, SA may have the potential to impede the vicious cycle of oxidative stress, neuro-inflammation and cholinergic deficiency and can thus put forward a new option for the treatment of AD. Hence, in the present study, we hypothesized that SA may exhibit neuro-protection in ICV-STZ induced cognitive impairment. Behavioral parameters for spatial memory and memory consolidation, biochemical estimations of oxidative biomarkers, pro-inflammatory cytokines like IL-1β and TNF-α, expression of ChAT and neuron loss in the hippocampus were assessed in order to provide insight into the neuro-protective effects of SA in ICV-STZ induced sporadic AD model in rats.

## 2. Materials and Methods

### 2.1. Animals

Wistar rats (250–300 g, male) were obtained from the Central Animal Facility after necessary approval (991/IAEC/2017) from the Institutional Animal Ethics Committee, All India Institute of Medical Science, New Delhi, India. Committee for the Purpose of Control and Supervision of Experiments on Animals (CPCSEA) protocols were followed for animal care and experiments. Animals were housed in the animal acclimatization facility under standard laboratory settings with natural dark/light cycle. The rats were kept in cages (polyacrylic, 38 × 23 × 10 cm) with maximum of 4 animals per cage, fed with standard feed and tap water *ad libitum*.

### 2.2. Drugs and Chemicals

Streptozotocin, sinapic acid, 5′5-dithiobis (2-nitrobenzoic acid) (DTNB), sodium dodecyl sulphate, tetramethoxy propane, thio-barbituric acid, dithiothreitol (DTT), protease inhibitor and phenylmethylsulphonylfluride (PMSF) were procured form Sigma Aldrich, USA. Tween-80 was purchased from Merck, Germany. ELISA kits for TNF-α and IL-1β were purchased from Diaclone (France). Primary antibodies against ChAT and goat anti-rabbit secondary antibodies were obtained from Abcam Plc, UK. The primary antibody against neurons was procured from Cell Signaling Technology, USA.

### 2.3. ICV-STZ Injection

After baseline behavioral assessment, rats were anaesthetized using chloral hydrate (300 mg/kg, intra-peritoneal route). The dorsal surface of the skull was shaved; povidone-iodine solution was applied, followed by 70% alcohol. Animals were placed in a stereotactic frame. After a midline sagittal incision, the skull was exposed to reach bregma. Burr holes were made bilaterally, over lateral ventricles, with the help of a dental micro-motor machine using the coordinates of 0.8 mm posterior to bregma, 1.5 mm lateral to sagittal suture, and 3.6 mm ventral to dura, as described earlier [25]. STZ was prepared fresh by dissolving in normal saline immediately before administration. With the help of a Hamilton syringe, STZ was administered at a dose of 3 mg/kg bilaterally on day 1 [26]. To ensure dispersal of STZ, the Hamilton syringe was left in place for few minutes following the injection. After surgery, rats were kept in individual cages to recover from anesthesia and two rats per cage were housed.

### 2.4. Drug Treatment

The rats were divided into the following groups; (i) sham (all surgical procedures except STZ administration), (ii) STZ (3 mg/kg, on day 1), (iii) SA 10 (STZ at day 1 + SA 10 mg/kg, p.o for 21 days), (iv) SA 20 (STZ at day 1 + SA 20 mg/kg, p.o for 21 days), and (v) Per se (only SA 20 mg/kg for 21 days). SA was dissolved in 20% tween-80 and administered once daily for 21 days after ICV-STZ injection and same volume of vehicle was given to sham and STZ groups. Neurobehavioral parameters were assessed on day 0 and 21 days after ICV injection of STZ. The animals were sacrificed on day 21. The brain samples were collected and stored at −80 °C for biochemical estimation (Figure 1).

### 2.5. Behavioural Assessment


*Neuro-behavioral Tests*


For assessment of cognitive impairment, the well accepted Morris water maze test (to assess spatial memory) and passive avoidance paradigm (to assess maintenance of short term memory) were performed on day 0 and 21 days after STZ injection.


*Morris Water Maze Test*


To test the spatial memory, the Morris water maze test was performed [9,27]. The Morris water tank was divided into 4 quadrants (1, 2, 3 and 4) virtually with a platform in the 4th quadrant. Each animal received four trials for four days of 120 s each and, on the fifth day, a retention trial was performed. During the retention trial, the platform was taken out and the duration was reduced to 60 s. The Morris water maze retention trial was performed on day 0 and 21 days after ICV-STZ injection. The latency to reach the platform (escape latency) and time spent in the 4th quadrant was recorded and analyzed using the Any-maze video tracking system.


*Passive Avoidance Test*


Passive avoidance apparatus was used to test memory retention deficit [9]. Briefly, the rat was placed in the lighted chamber during the acquisition trial. The guillotine door was opened after 30 s of habituation. The time the rat took to enter the dark chamber, called initial latency, was noted. Rats showing an initial latency of >60 s were omitted. Immediately, as the guillotine door was closed, a foot shock (75 V, 0.2 mA, 50 Hz) was given through the grid floor for 2 s. The rats were returned to the home cage 5 s later. On the next day, retention latency was noted in the same manner without delivering foot shock up to a maximum of 300 s.

### 2.6. Tissue Collection

On day 21, rats were decapitated under sodium pentobarbital (150 mg/kg) anesthesia and the brains were quickly removed. Brains were dissected into hippocampus and cerebral cortex, over an ice cold glass plate and stored at −80 °C until further analysis. For estimation of oxidative stress parameters and pro-inflammatory markers, tissues were homogenized in 8× (*w*/*v*) ice–cold 0.1 M phosphate buffer (pH 7.4). For ChAT expression study using western blot, the tissues were homogenized in lysis buffer. Bradford’s assay was used to estimate the total amount of protein. For immunofluorescence study, rats were euthanized on the 21st day, and were transcardially perfused with normal saline and with 4% paraformaldehyde. The brains were removed, and kept in 4% paraformaldehyde for 24 h followed by cryo-protection in sucrose gradients. OCT blocks were prepared and 14 μm thick coronal sections of the hippocampus region were taken on gelatin coated slides.


**Estimation of Reduced Glutathione (GSH)**


GSH levels in the brain were estimated as described by Ellman [28]. Briefly, homogenates were centrifuged with 5% trichloroacetic acid. To the supernatant, 100 μL phosphate buffer (pH-8.4) and 0.5 mL DTNB were added. The absorbance was read at 412 nm within 15 min. Glutathione was used as a standard. The concentration of GSH was determined and reported as μg/mg of protein.


**Measurement for Lipid Peroxidation (MDA)**


Malondialdehyde estimation was done by the method described by Ohkawa et al. [29]. Briefly, acetic acid 0.75 mL (20%) pH-3.5, 0.75 mL (0.8%) thiobarbituric acid and 0.1 mL (8.1%) sodium lauryl sulphate were added to 0.05 mL of tissue homogenate and kept at 100 °C for 1h in water bath. After cooling, 2.5 mL of n-butanol:pyridine followed by 0.5 mL distilled water were added. The organic layer was separated by centrifuge and absorbance was measured at 532 nm. Tetramethoxy propane was used as MDA standard and the MDA amount was presented as nmol/mg of protein.


**TNF-α and IL-1β**


Levels of TNF-α and IL-1β were estimated in the cortex and hippocampus using ELISA kits from Diaclone (France). Monoclonal antibodies specific to TNF-α and IL-1β were pre-coated onto the wells of the microtiter plates provided. Standards and samples were added to microtiter wells. After washing, biotinylated antibody was added and incubated. Wells were again washed, streptavidin-HRP was added. Substrate solution was added to induce a blue coloured reaction product that turned yellow after adding stop solution. Absorbance was measured at 450 nm and cytokines levels were calculated and reported as pg/mg of protein.


**ChAT Expression by Western Blot**


Equal amounts of protein of each denatured sample were subjected to electrophoresis on 10% SDS-PAGE. Consecutively, resolved proteins were then transferred onto a nitrocellulose membrane. Membrane was blocked using 5% non-fat milk. After overnight incubation with primary antibody to ChAT (1:1000, rabbit polyclonal anti-ChAT, ab178850) and beta actin (1:3000, rabbit polyclonal anti-beta actin, ab8227), membrane was incubated with secondary antibody (1:4000, Goat-anti-rabbit IgG-H&L HRP, ab6721) after two washes with PBST buffer. Finally, chemiluminescence reagent was used to visualize the membrane. A gel documentation system (FluorChem E, Cell Biosciences, Santa Clara, CA, USA) was used to analyze the optical densities of bands.

### 2.7. Neuronal Loss by Immunofluorescence

Coronal sections were washed in PBS, blocked with 10% normal goat serum followed by permeabilization with 0.2% triton-X100 in PBS. Sections were incubated with primary antibody for NeuN (1:100, mouse monoclonal anti-NeuN, Cell Signaling Technology, #94403S) for 2 h followed by incubation with the corresponding secondary antibody, goat anti-mouse IgG H&L (Phycoerythrin, ab97024), for 1 h. Finally, the sections were washed, counterstained with DAPI and mounted. Immunofluorescence images from the CA1 region of the hippocampus were acquired using a Nikon-Ti microscope with the help of NIS-Element software.

### 2.8. Statistical Analysis

Statistical analysis was done by using Graph Pad prism 5. Data were expressed as mean ± SEM. One-way analysis of variance followed by Bonferroni post hoc analysis was used to test significance. A *p*-value of less than 0.05 was considered statistically significant. 

## 3. Results

### 3.1. Effect of Sinapic Acid on Morris Water Maze Test

Baseline escape latency was similar among the groups in the probe trial. One way ANOVA with Bonferroni multiple comparison revealed a significant (*p* < 0.001) increase in the escape latency in the STZ group on day 21 when compared with the sham group, indicating memory deficits. Sinapic acid (20 mg/kg) significantly reduced the STZ-induced increased escape latency. In the per se group, there was no significant difference in escape latency as compared to sham (Figure 2).

At baseline (before STZ administration, day 0) during the probe trial, the time spent in the fourth quadrant was similar among different groups (Figure 3). One way ANOVA followed by Bonferroni’s multiple comparison post hoc analysis revealed a significant (*p* < 0.01) reduction in time spent in the fourth quadrant on day 21 in the STZ group as compared to sham. SA (20 mg/kg) significantly normalized the STZ-induced reduction in time spent in the fourth quadrant on day 21 (*p* < 0.05). However, SA acid (10 mg/kg) did not produce significant improvement in time spent in the target quadrant when compared with STZ. Further, there was no significant difference in time spent in the fourth quadrant in the per se group of SA as compared with the sham group.

### 3.2. Effect of Sinapic Acid on Passive Avoidance Test

At baseline, no difference in retention latency was observed among the groups. One way analysis of variance followed by Bonferroni post hoc analysis showed a significant (*p* < 0.001) reduction in retention latency in the STZ group on day 21 when compared to the sham group. Treatment with SA increased the STZ-induced decreased retention latency. The difference was significant with both 10 and 20 mg/kg doses on day 21. The retention latency of the per se group was similar to that of the sham group (Figure 4).

### 3.3. Effect of Sinapic Acid on Oxidative Stress Parameters


**Effect of Sinapic Acid on Brain GSH Levels**


There was a significant (*p* < 0.001) reduction in the levels of GSH in the STZ group when compared with the sham group (21.9 ± 2.2 vs. 7.9 ± 1.0 μg/mg of protein in the cortex; 22.9 ± 2.9 vs. 6.2 ± 0.8 μg/mg of protein *p* < 0.001 in the hippocampus). SA significantly attenuated the STZ-induced reduced levels of GSH at a dose of 20 mg/kg both in the cortex (*p* < 0.05) as well as in the hippocampus (*p* < 0.01) when compared to the STZ group (Figure 5). The per se group of SA did not show any significant difference in the GSH levels as compared to the sham group.


**Effect of Sinapic Acid on Brain MDA Levels**


As compared to the sham group, levels of MDA were significantly (*p* < 0.001) increased from 43.7 ± 4.5 to 95.1 ± 5.7 nmol/mg of protein in the cerebral cortex and from 42.5 ± 6.6 to 82.6 ± 5.9 nmol/mg of protein in the hippocampus of the STZ group. SA 20 mg/kg significantly prevented STZ-induced raised MDA levels in the cortex (*p* < 0.001) and hippocampus (*p* < 0.05). SA 10 mg/kg also significantly reduced MDA levels in the cortex (*p* < 0.05), whereas in the hippocampus the effect was not significant. There was no difference in the per se group as compared to the sham group (Figure 6). 

### 3.4. Effect of Sinapic Acid on Levels of Inflammatory Cytokines


**Brain TNF-α Level**


STZ administration caused significant increase in TNF-α levels (92.3 ± 14.8 vs. 159.4 ± 14.5 pg/mg of protein, *p* < 0.01 and 88.8 ± 10.4 vs. 172.5 ± 15.1 pg/mg of protein; *p* < 0.001, respectively, in the cerebral cortex and hippocampus). As compared to STZ group, SA 20 mg/kg significantly reduced TNF-α levels in the cortex and hippocampus. TNF-ɑ levels of the per se group were similar to those of the sham group (Figure 7).


**Brain IL-1 β Levels**


A significant increase in cortical and hippocampal IL-1β levels in the STZ group was observed in both the cortex and hippocampus (90.3 ± 7.4 vs. 199.4 ± 10.1 pg/mL of protein; *p* < 0.001 and 85.1 ± 7.4 vs. 170.46 ± 10.1 pg/mL of protein; *p* < 0.001) as compared to the sham group. SA 20 mg/kg significantly reduced the IL-1β level (129 ± 15.3 pg/mL and 106 ± 8.3 pg/mL of protein; *p* < 0.01) in the cortex and hippocampus. However, SA per se did not show any difference in IL-1 β when compared to sham (Figure 8).

### 3.5. Effect of Sinapic Acid on Expression of ChAT

One way ANOVA followed by Bonferroni post hoc analysis showed significantly (*p* < 0.01) reduced expression of hippocampal ChAT in the STZ group as compared to the sham group. In the cortex, as compared to sham, there was no significant change in ChAT expression in the STZ group. In the hippocampus, SA 20 mg/kg treatment significantly (*p* < 0.05) normalized the STZ-induced decreased expression of ChAT when compared to the STZ group (Figure 9). However, the SA per se group did not show any difference in ChAT expression when compared to sham.

### 3.6. Effect of Sinapic Acid on Neuron Loss

To check the effect of sinapic acid on neuronal loss in the CA1 region, brain hippocampal slices were stained with NeuN. The STZ group showed significant neuronal loss (*p* < 0.01) in the CA1 region of the hippocampus as compared to sham (Figure 10). However, the treatment with sinapic acid significantly (*p* < 0.05) attenuated the STZ-induced neuronal loss as compared to the STZ group. 

## 4. Discussion

The results of the present study revealed that intracerebroventricular injection of STZ mimicked AD like behavioral changes including decreased time spent in the fourth quadrant, enhanced escape latency in the Morris water maze test, decreased retention latency in the passive avoidance test, impaired oxidative status accompanied by increased pro-inflammatory cytokines and decreased ChAT expression in the hippocampus. Administration of sinapic acid for 21 days attenuated the STZ-induced changes in behavioral and oxidative parameters, inflammatory cytokines and levels of ChAT. 

The current therapies for AD primarily focus on restoring cholinergic transmission or inhibiting NMDA receptor and amelioration of symptoms [30]. Since oxidative stress and neuro-inflammation are early features of AD pathogenesis, targeting them and replenishing the cholinergic transmission may provide a novel therapy in AD [19]. In the recent past, polyphenolic compounds have been explored extensively because of their antioxidant properties [31]. SA is a hydroxycinnamic acid compound with proven anti-oxidant and anti-inflammatory properties [22,32]. Hence, the current study was planned to evaluate the neuro-protective effect of SA in ICV-STZ-induced cognitive impairment in rats.

ICV-STZ is a well-accepted experimental model to mimic sporadic AD. In the present study, bilateral ICV-STZ injection induced cognitive impairments, as shown by behavioral and pathological alterations, mimicking sporadic AD in rats [13,33,34,35].

Further, the brain has greater metabolic activity, high content of polyunsaturated fatty acids, less glutathione and high oxygen consumption, making it explicitly available to free radical attack [36,37]. Imbalance between the production of reactive oxidizing metabolites and their elimination by the enzymatic or non-enzymatic antioxidant system causes the state of oxidative stress [38]. An excess of free radicals causes damage to the membrane’s lipids and proteins, which eventually leads to a decrease in GSH and an increase in MDA levels in the brain [34]. Moreover, MDA and aldehydes are associated with neuronal death by modulating the ATPases associated with ionic transfers as well as calcium homeostasis [39]. Dysregulation of calcium homeostasis, an intrinsic ionic regulator, plays a critical role in the initiation and progression of AD as the calcium binding protein calmodulin, and the calmodulin binding protein neurogranin, are reported to be involved in AD associated long-term depression. Calmodulin inhibitors improved cognitive functions in an animal model of vascular dementia [40,41]. Further, disturbed calcium homeostasis induced increased intracellular calcium and calmodulin leads to activation of NMDA and AMPA receptors, responsible for long term depression associated with AD [40]. Glutamate receptor over-activation is also responsible for ChAT inhibition [42] which was completely prevented by NMDA and AMPA receptor antagonists [43]. Taken together, these events ultimately lead to a state of neuroinflammation and cholinergic dysfunction.

STZ-induced cognitive impairment was assessed by behavioral tests. Morris water maze and passive avoidance paradigms are well accepted behavioral tests used in experimental AD research. The widespread conformational validity of the Morris water maze as a measure of spatial navigation and predictive memory, and use of the passive avoidance test, which assesses the basic capability to learn and remember the existence of a shock stimulus, with the advantages of minimal practice and quick learning make both of these behavioral tests suitable to assess cognitive decline in a sporadic AD model [44]. The escape latency or the time required to locate the platform zone was more, whereas the time spent in the target quadrant was less in STZ injected rats, indicating memory impairment. The results suggest that SA improved cognitive impairment as shown by the shorter escape latency and longer time spent in the fourth quadrant. Similarly, in the passive avoidance test, there was reduced retention latency in the STZ group and SA significantly attenuated the reduced retention latency. Hence, SA improved cognitive impairment as evidenced by the improvement in behavioral test performance.

In AD, cognitive impairment is associated with reduced antioxidants, high levels of oxidized proteins and lipid peroxidation end products, and the formation of free radicals such as peroxides, aldehydes and oxidative modifications in mitochondria [45]. Excessive oxidative stress induces mitochondrial impairment and causes a decrease in ATP production, altered calcium homeostasis and a further increase in reactive oxygen species production. These events lead to DNA, lipid and protein damage and long-term increase can cause mitochondrial DNA mutation, ultimately progressing the disease severity [46] Moreover, this oxidative stress may aggravate a vicious and interminable cycle of neuro-inflammation, which further promotes oxidative stress [47,48]. Previous studies also substantiate ICV-STZ-induced decreased GSH and increased MDA levels in the cortex and hippocampus [8]. The results of our study are in accordance with the above mentioned studies wherein SA treatment significantly counteracted the altered levels of oxidative stress markers. This effect may be attributed to the free radical scavenging ability and phenoxyl hydrogen atom donating capacity of SA, leading to the neutralization of free radicals to form phenoxyl radicals [21]. Neuro-inflammation, as reflected by astrogliosis and microglial activation, is another prominent feature contributing to AD pathogenesis [49,50]. The pro-inflammatory cytokines TNF-α and IL-1β play crucial roles in AD neuro-inflammation [51]. In previous studies along with studies from our laboratory, the levels of TNF-α and IL-1β were increased in the cortex and hippocampus after STZ injection [25,52]. In agreement with the above results, in the present study as well, the levels of both pro-inflammatory cytokines were significantly increased in the STZ group as compared to the sham group. SA treatment significantly reduced the levels of TNF-α and IL-1β in the cortex and hippocampus. The anti-inflammatory properties of SA are most likely be mediated by its ability to suppress NF-κB p65 activation, which further reduces pro-inflammatory cytokines [24,53]. Antioxidant effects and anti-inflammatory activity of SA may be responsible for the retrieval in cognitive functions. 

Acetylcholine, a major cholinergic neurotransmitter in the brain, plays an important role in learning and memory. Its physiological action is terminated by acetylcholinesterase (AChE) [54]. Notably, choline acetyltransferase (ChAT), the rate limiting enzyme in ACh formation, catalyzes the transfer of an acetyl group from acetyl-CoA to choline [55]. Henceforth, depletion of ChAT activity in the hippocampus can be correlated with the severity of cognitive disturbance in AD [56,57]. It has been reported that levels of ChAT decrease in Alzheimer’s patients with severe cognitive impairment and the same has been observed in ICV-STZ-induced cognitive impairment [58]. Likewise, in the present study, in the STZ group, there was reduced ChAT expression. SA normalized the ChAT expression and mitigated the cognitive impairment in rats. Similar effects of SA have been reported, where its derivative sinapine restored the levels of acetylcholine and improved learning and memory [20,23,59]. Furthermore, studies have revealed that ICV-STZ administration has been associated with neuronal loss in the CA1 region of the hippocampus [60,61,62]. We found that apart from improvements in behavioral parameters, oxidative stress, anti-inflammatory cytokines and hippocampal ChAT expression, SA treatment was accompanied by attenuation of STZ-induced neuronal loss in the CA1 region of the hippocampus. Although in the future, in-depth mechanistic studies about the role of SA-induced neuro-protection are needed, our results show a role of SA in the attenuation of STZ-induced cognitive impairment in rats.

## 5. Conclusions

In the present study, our findings showed that ICV-STZ-induced impairment of behavioral parameters; increased oxidative stress and increased pro-inflammatory cytokines were reversed by SA treatment for 21 days. Free radical scavenging and anti-inflammatory activities of SA may be attributed to these effects. SA attenuated the cognitive impairment by restoring the acetylcholine levels through up-regulation of ChAT protein expression. AD being a multifactorial disease, SA can target multiple mechanistic pathways of AD such as oxidative stress, neuro-inflammation and cholinergic dysfunction. These results thus indicate a therapeutic potential of SA in AD.

## Figures and Tables

**Figure 1 brainsci-10-00923-f001:**
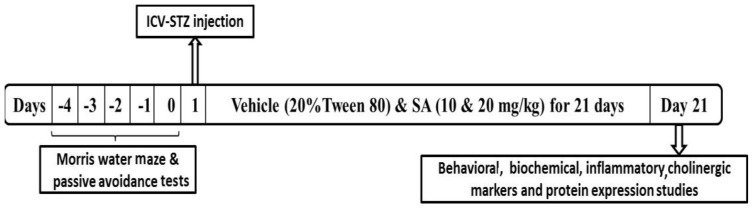
Schematic diagram showing experimental protocol. Morris water maze training trials were given for 4 days (−4 to −1 day) followed by retention trial on day 0. Intracerebroventricular streptozotocin was administered on day 1. Experimental rats were dosed with sinapic acid for the next 21 days. On day 21, rats were sacrificed after behavioral assessment.

**Figure 2 brainsci-10-00923-f002:**
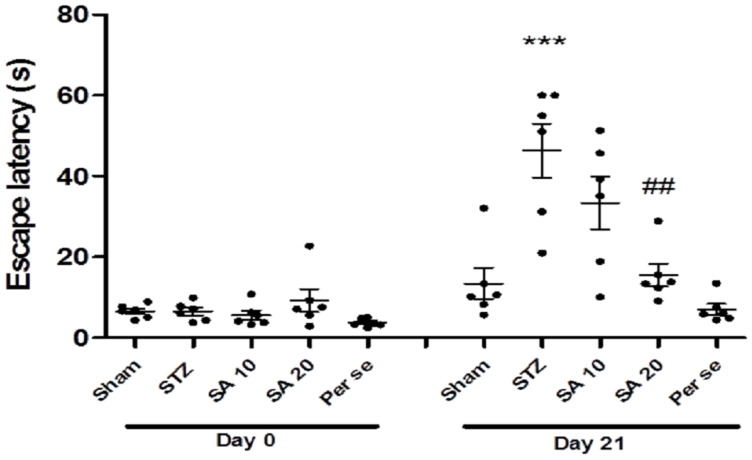
Effect of sinapic acid on latency to reach the platform zone. Data are presented as mean ± SEM, (*n* = 6). * as compared to the sham group, # as compared to the STZ group. *** *p* < 0.001, ## *p* < 0.01. STZ—Streptozotocin, SA—Sinapic acid.

**Figure 3 brainsci-10-00923-f003:**
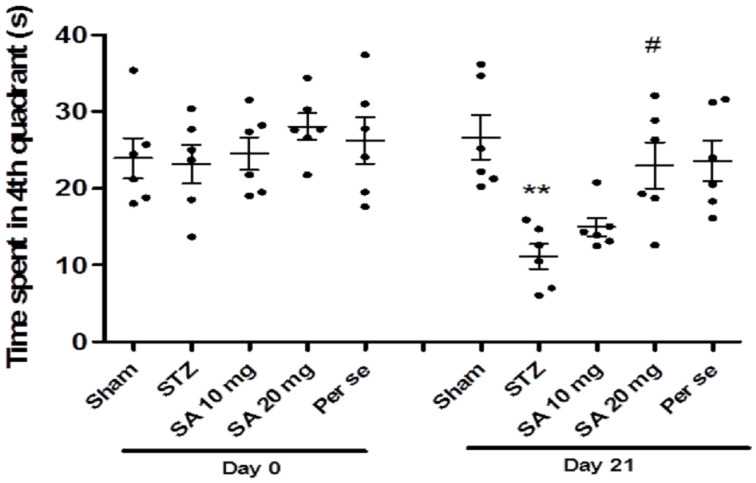
Time spent in the target quadrant in the Morris water maze test. Data are presented as mean ± SEM (*n* = 6). * as compared to the sham group, # as compared to the STZ group. ** *p* < 0.01, # *p* < 0.05. STZ—Streptozotocin, SA—Sinapic acid.

**Figure 4 brainsci-10-00923-f004:**
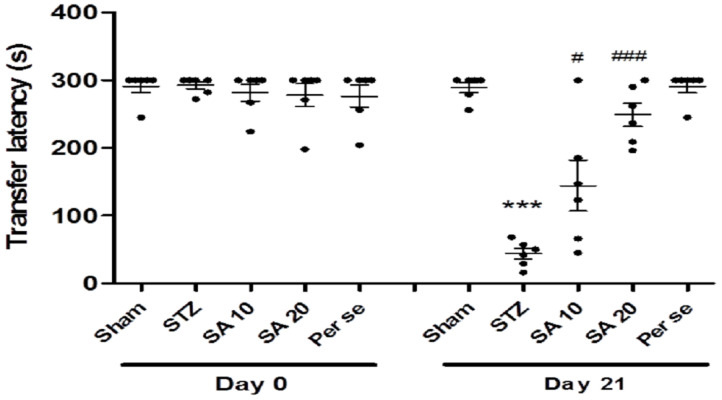
Effect of different doses of sinapic acid on retention latency in passive avoidance. Data are presented as mean ± SEM, (*n* = 6), * as compared to the sham group, # as compared to the STZ group. *** *p* < 0.001, # *p* < 0.05, ### *p* < 0.001. STZ—Streptozotocin, SA—Sinapic acid.

**Figure 5 brainsci-10-00923-f005:**
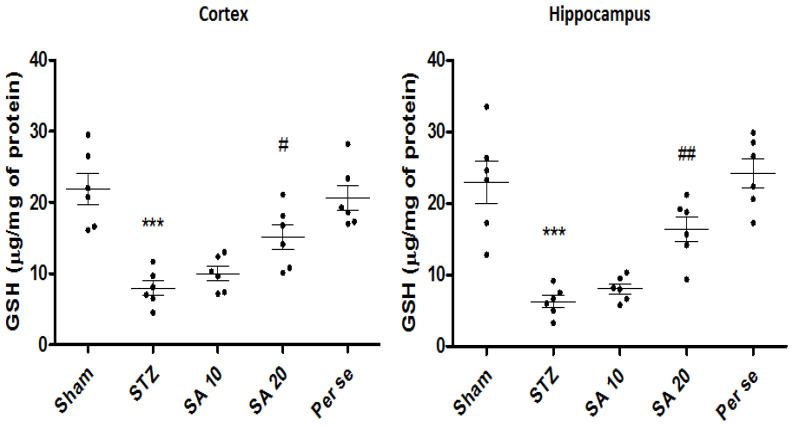
Effect of sinapic acid on brain GSH levels in the cortex and hippocampus. Data are presented as mean ± SEM, (*n* = 6). * as compared to the sham group, # as compared to the STZ group. *** *p* < 0.001, # *p* < 0.05, ## *p* < 0.01. STZ—Streptozotocin, SA—Sinapic acid.

**Figure 6 brainsci-10-00923-f006:**
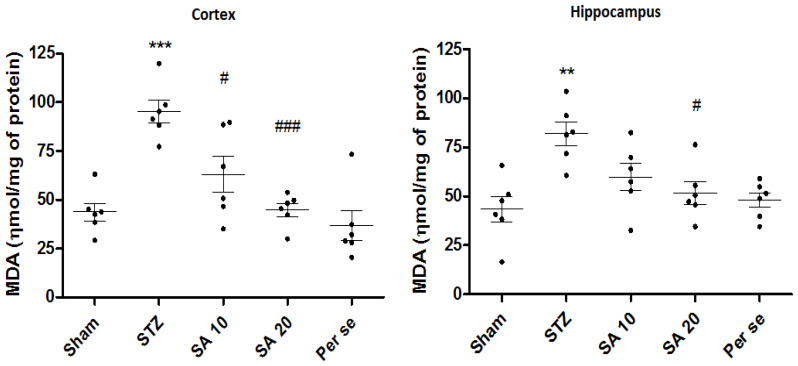
Effect of sinapic acid on MDA levels in the cortex and hippocampus. Data are presented as mean ± SEM, (*n* = 6). * as compared to the sham group, # as compared to the STZ group. ** *p* < 0.01, *** *p* < 0.001, # *p* < 0.05, ### *p* < 0.001. STZ—Streptozotocin, SA—Sinapic acid.

**Figure 7 brainsci-10-00923-f007:**
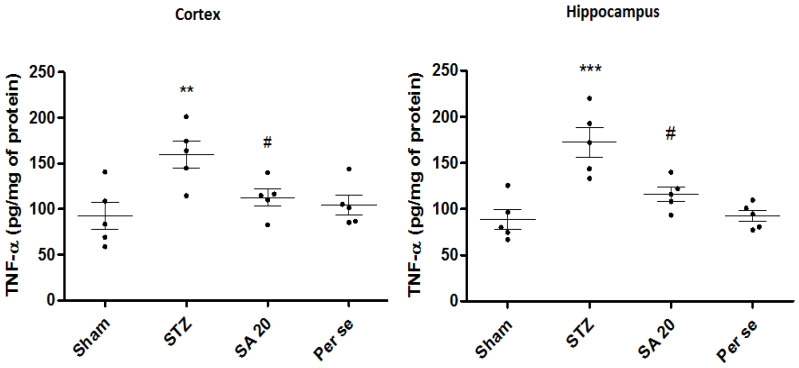
Effect of sinapic acid on TNF-α level in the cortex and hippocampus. Data are presented as mean ± SEM, (*n* = 5). * as compared to the sham group, # as STZ compared to the STZ group. ** *p* < 0.01, *** *p* < 0.001, # *p* < 0.05. STZ—Streptozotocin, SA—Sinapic acid.

**Figure 8 brainsci-10-00923-f008:**
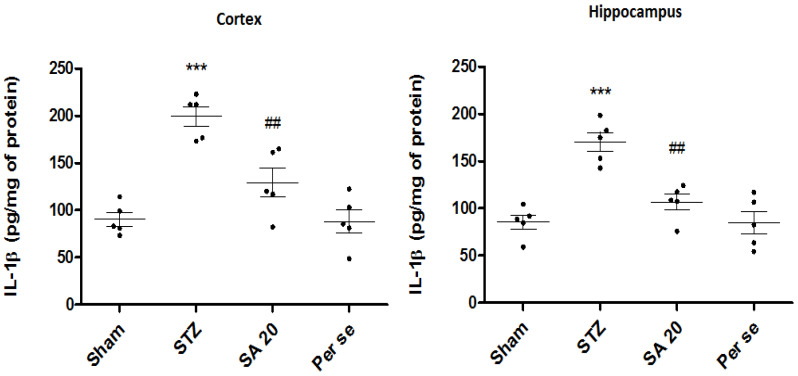
Effect of sinapic acid on IL-1β levels in the cortex and hippocampus. Data are presented as mean ± SEM, (*n* = 5). * as compared to the sham group, # as compared to the STZ group. *** *p* < 0.001, ## *p* < 0.01. STZ—Streptozotocin, SA—Sinapic acid.

**Figure 9 brainsci-10-00923-f009:**
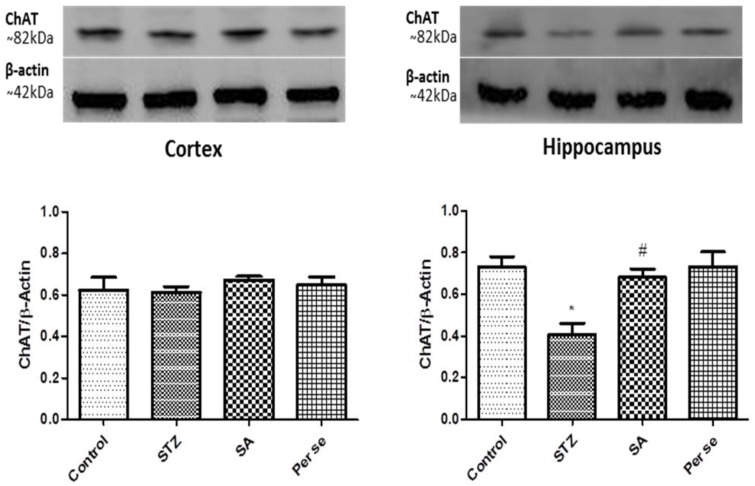
Effect of SA on expression of ChAT in the cortex and hippocampus. Representative blots showing expression of ChAT in the cortex and hippocampus. Bar diagrams show change in ChAT expression when normalized against β-actin. Data are presented as mean ± SEM, (*n* = 4). * as compared to the sham group, # as compared to the STZ group. * *p* < 0.05, # *p* < 0.05. STZ—Streptozotocin, SA—Sinapic acid.

**Figure 10 brainsci-10-00923-f010:**
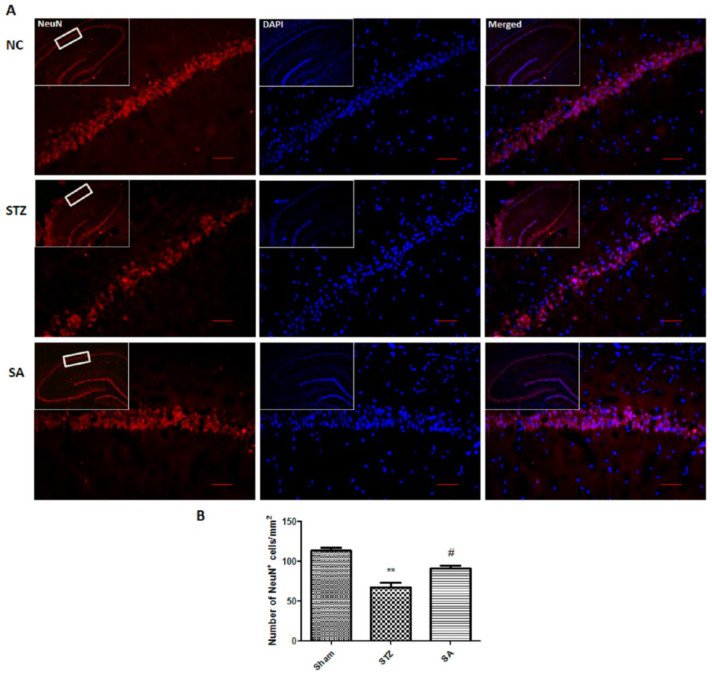
Effect of SA on STZ-induced neuronal loss. (**A**). Representative photomicrographs showing the effect of SA on neuronal loss in the CA1 region of the hippocampus (scale bar = 100 µm). (**B**). The bar diagram revealed a significant increase in the number of surviving neurons in the SA group as compared to the STZ group. Data are presented as mean ± SEM, (*n* = 3). * as compared to the sham group, # as compared to the STZ group. ** *p* < 0.01, # *p* < 0.05. STZ—Streptozotocin, SA—Sinapic acid.

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
