# Peer review of "Sinapic Acid Alleviates Oxidative Stress and Neuro-Inflammatory Changes in Sporadic Model of Alzheimer’s Disease in Rats"

_brainsci, 2020, doi:10.3390/brainsci10120923_

Round 1
Reviewer 1 Report
Comments to Verma et al.:
In this paper, Verma et al. evaluated the neuro-protective effect of sinapic acid (SA) on intracerebroventricular streptozotocin (ICV-STZ) induced AD rats, where oxidative stress, neuro-inflammation and neuronal loss were partially rescued in cortex and hippocampus. Biochemical analysis are sounds and controls are solid.
Major:
- Since you measured TNF-a and IL-1b levels by ELISA in figure 7 and 8, please confirm this data with western blots. Please also include other known inflammatory cytokines as measured by qRT-PCR, this will give a wide effect of SA in inflammatory response.
- Figure 9, no major difference of ChAT expression detected by western blot in cortex, this could arise from the minor changes in the whole cerebral cortex. Please repeat this with immunofluorescence staining to see any changes of ChAT at specific cortical brain regions.
- Does ICV_STZ model induce gliosis in the brain? If yes, please analysis expression of microglia and astrocyte in response to STZ and SA.
Minor:
- Please change bar graphs into dot blots, this will give a better visualization of data variations.
- It’s a bit weird no effect of SA per se detected in all the parameters analyzed.
- Please explain why you used 6 animals in behavior, but used 5 mice in the biochemical experiments.
Author Response
Dear Reviewer,
Thank you very much for the constructive comments. Most of the concerns have been incorporated in the revised manuscript. These have improved our manuscript. Point wise clarifications are provided in the attached file. Please see the attachment.

Reviewer 2 Report
Dear Authors,
The authors studied the neuroprotective effects of a hydroxycinnamic acid derivative sinapic acid (SA) on the intracerebroventricular streptozotocin (STZ)-induced cognitive impairment in rats. Administration of intragastric SA (20 mg/kg) for three weeks significantly improved cognitive impairment assessed by Morris water maze and passive avoidance tests, altered the levels of reduced glutathione, malondialdehyde, tumor necrosis factor alpha, and interleukin 1 beta in cortex and hippocampus, and significantly improved STZ-induced decreased expression of choline acetyltransferase by western blot and neuronal loss by immunofluorescence microscopy. The authors concluded that SA exhibited neuroprotective activity against STZ-induced oxidative stress, neuro-inflammation, cholinergic dysfunction, and neuronal loss, all of which contributed to improved cognitive functions.
Please consider the following parts:
- Page 1, Line 19: “ChAT”, Please define the abbreviation.
- Page1, Line 40: “SAD”, Please define the abbreviation.
- Page 2, Line 43: Some of the references are outdated. Please update it. Suggested reference for multiple factors for neurodegenerative diseases: Tanaka, M.; Toldi, J.; Vécsei, L. Exploring the Etiological Links behind Neurodegenerative Diseases: Inflammatory Cytokines and Bioactive Kynurenines. J. Mol. Sci.2020, 21, 2431.
- Page 3, Line 113: There are several animal models for cognitive function. Please present the rationale here to choose the tests.
- Page 5, Figure 2 and other figures: Please show p-value for #. ” ##p<0.01”?
- Page 6, Line 222 and afterwards: What are the symbols after the bullet?
- Page 11, Lines 301-304: A role of N-methyl-D-aspartate (NMDA) receptor and its antagonist in Alzheimer’s disease (AD) is missing. Recommended reference: Tanaka, M.; Bohár, Z.; Vécsei, L. Are Kynurenines Accomplices or Principal Villains in Dementia? Maintenance of Kynurenine Metabolism. Molecules2020, 25, 564.
- Page 11, Line 308: Please choose either” sporadic AD” or “SAD”.
- Page 11, Lines 311-317: Please discuss oxidative stress, inflammation, and antioxidants capacity in neurodegenerative diseases. Suggested reference: Tanaka, M.; Vécsei, L. Monitoring the Redox Status in Multiple Sclerosis. Biomedicines2020, 8, 406.
It also deserves to describe more about calcium homeostasis. Suggested reference: O’Day, D.H. Calmodulin Binding Proteins and Alzheimer’s Disease: Biomarkers, Regulatory Enzymes and Receptors That Are Regulated by Calmodulin. Int. J. Mol. Sci. 2020, 21, 7344.
- Pages 11, Lines 326-331: It deserves to discuss recent findings of roles of mitochondria and oxidative stress. Suggested references: Brito, L.M.; Ribeiro-dos-Santos, Â.; Vidal, A.F.; de Araújo, G.S., on behalf of the Alzheimer’s Disease Neuroimaging Initiative; Differential Expression and miRNA–Gene Interactions in Early and Late Mild Cognitive Impairment. Biology2020, 9, 251. Catanesi, M.; d’Angelo, M.; Tupone, M.G.; Benedetti, E.; Giordano, A.; Castelli, V.; Cimini, A. MicroRNAs Dysregulation and Mitochondrial Dysfunction in Neurodegenerative Diseases. J. Mol. Sci.2020, 21, 5986.
- Pages 11-12, Lines 348-364: Discussion on the glutamatergic nervous system is missing. This study did not investigate the NMDA receptors, but it deserves to discuss it in relation to oxidative stress, inflammation, and the cholinergic nervous system.
- Please correct typological errors, especially space, and period in References.
- Please follow the citation format of Brain Sciences: [X]
The manuscript contains ten figures and 53 references. I recommend including more references, preferably at least up to 75. The manuscript is of great value investigating a beneficial effect of SA in an animal model of sporadic AD and its mechanisms including antioxidant, anti-inflammatory, pro-cholinergic, and neuroprotective effects. I recommend this manuscript for publication after minor revision.
I declare no conflict of interest regarding this manuscript.
Author Response
Dear Reviewer,
Thank you very much for the detailed review and the constructive comments. We could not repeat the parameters suggested by you due to financial and time constraints. Point wise clarifications are provided in the attached document. Please see the attachment.
